# Aligning with Patients and Families: Exploring Youth and Caregiver Openness to Pediatric Headache Interventions

**DOI:** 10.3390/children9121956

**Published:** 2022-12-13

**Authors:** Allison M. Smith, Zoë J. Schefter, Hannah Rogan

**Affiliations:** 1Division of Pain Medicine, Department of Anesthesiology, Perioperative & Pain Medicine, Boston Children’s Hospital, Waltham, MA 02453, USA; 2Division of Psychology, Department of Psychiatry, Harvard Medical School, Boston, MA 02115, USA

**Keywords:** pediatric pain, pediatric headache, openness, parent–child concordance

## Abstract

Primary headache disorders are common yet underestimated in youth, resulting in functional disability, decreased quality of life, and caregiver burden. Despite the ubiquity of options, adherence remains challenging for families. One potential factor impacting willingness to engage in recommended treatments is openness. This study explored openness to multidisciplinary headache interventions and the relationships with demographic, pain-related, and psychological variables, among youth and their caregivers. Participants (*n* = 1087) were youth/caregiver dyads presenting for initial headache evaluation. They completed assessments of openness to headache treatments, medical information, functional disability, and pain-related distress. Overall openness was moderately high for youth and caregivers, and highly correlated between them (r = 0.70). Relationships between youth/caregiver openness to specific interventions were moderate–high (r = 0.42–0.73). These were stronger for interventional techniques but weaker for lifestyle changes. In hierarchical regression models predicting youth and caregiver openness, we found that counterpart openness accounted for the largest portion of variance in their own openness (31–32%), beyond demographic (3%), pain-related (10%), and psychological variables (2–3%). Our findings highlight the importance of involving caregivers in pediatric headache management, given their influence on youth openness and potential involvement in adherence. Awareness of youth/caregiver openness may guide clinicians providing recommendations.

## 1. Introduction

Pediatric headache is widespread, with community prevalence estimates over periods of one month to lifetime in youth averaging nearly 60% [1,2] and a subset of these (~12%) going on to develop chronic headaches [3]. Chronic headaches are also emotionally and financially costly, resulting in significant functional disability [4,5], and economic burden to families [6]. It is well-known that all chronic pain experiences, including chronic headache, entail complex interactions between biological, psychological, and social factors [7]. Because of the multifaceted nature of chronic headaches and variation in presentation, a wide array of treatment approaches is often considered. These can range from medical approaches (e.g., medication, interventional procedures, surgery), to lifestyle approaches (e.g., physical activity, nutrition/hydration, sleep hygiene interventions), and from psychological/biobehavioral approaches (e.g., cognitive-behavioral therapy (CBT), relaxation training, biofeedback) to complementary and alternative approaches (e.g., acupuncture, aromatherapy).

Despite the ubiquity of pediatric headache and the variety of treatment options, adherence to provider recommendations remains generally low [8]. Although research regarding adherence specifically in pediatric headache is scant, Simons and colleagues examined barriers to treatment engagement and patterns of non-adherence among pediatric patients across a variety of chronic pain presentations [9]. They found that, of those recommended to begin CBT, only 47% were fully adherent and, of those recommended to make a medication change, only 26% carried out this recommendation. Interestingly, physical therapy (PT) recommendations were the most likely to be carried through, with 100% of those advised to begin PT ultimately engaging in PT. Such patterns of non-adherence can be observed in subgroups within the pediatric chronic pain population as well. In their study of adherence in youth with inflammatory bowel disease, Varni and colleagues found that the presence of patient-perceived treatment barriers to medication adherence mediated the relationship between pain intensity and quality of life [10]. They concluded that such adherence barriers could be a prime target for future interventions to improve quality of life. Conversely, in a volunteer sample of pediatric chronic migraine patients, Kroon Van Diest and colleagues reported high adherence rates in their clinical trial [11]. Still, given the supports offered to research volunteers (e.g., reimbursement for time and travel expenses), it is difficult to generalize these findings to treatment-seeking patient populations. Thus, there is a clear need for better understanding of adherence in pediatric chronic headache samples. 

Given the detrimental impact of non-adherence not only to patient outcomes but also to quality of life, many researchers have focused on identifying the psychological barriers to adherence. In their study of adults in a tertiary headache care center, Matsuzawa and colleagues found that patient beliefs (e.g., perceptions of the recommended treatment, self-perceived readiness for change, appraisal of pain self-efficacy, pain treatment acceptance) may present psychological barriers to treatment adherence and thus impede recovery [12]. Similarly, in pediatric chronic headache, Simons and colleagues found that emotional and cognitive responses to pain, such as fear and catastrophizing, respectively, are influential factors in the pain experience, including willingness to engage in treatment [13]. Thus, again, underlying patient beliefs about pain appear to be an important component of chronic headache recovery. 

In youth with chronic pain, another critical factor that appears to influence patient beliefs and behaviors is caregiver modeling of pain attitudes, beliefs, and coping behaviors. Simons, Claar, and Logan explored parental responses to child pain and found a clear relationship between the nature of parental responses to the child’s pain and the child’s degree of disability and pain severity [14]. This highlights the importance of a caregiver modeling of adaptive pain beliefs and behaviors, given their impact on the child’s pain experience. Relatedly, Sieberg, Williams, and Simons found that parent distress and related protective responses in the context of their child’s pain may present barriers to treatment [15]. This suggests how influential parent beliefs and modeled behaviors are on their child’s functioning and recovery. 

One potentially relevant factor for understanding treatment adherence and willingness to engage in recommended pain interventions is openness to (i.e., the perceived acceptability of) various treatment approaches on the part of youth and their caregiver(s). Latifian and colleagues examined the role of openness, as a broad personality construct, in the recovery of adults with chronic pain [16]. They reported that being open to new experiences generally mediates relationships between pain perception and pain self-efficacy, pain management strategies, and resilience. Patients with greater openness were better able to manage their pain, both within the headache subgroup as well as among other pain subgroups. Claar and Scharf [17] also proposed that patient and youth perceptions of the effectiveness of various chronic pain interventions would likely influence treatment engagement and adherence, though they examined treatment modalities already being employed by their sample. 

Less is known about the factors that influence openness to or the perceived acceptability of interventions prior to engagement therein. Further, openness in youth with chronic headache is not well-understood, as it has yet to be studied in this population directly. This study seeks to bridge this gap in the literature by exploring patterns of self-reported openness to available interventions for chronic headaches among youth (and their caregivers) who are seeking multidisciplinary evaluation in a tertiary pediatric headache clinic. Specifically, this study examined relationships between youth and caregiver openness to numerous headache interventions and demographic factors, pain-specific factors, and psychological factors. Our examination of potential relationships with demographic factors was exploratory in nature. We hypothesized a direct relationship between pain intensity/frequency and openness to intervention in both youth and their caregivers, given that openness has been established as a relevant factor in chronic pain recovery in adults. We surmised that the more pain one perceives, the more open they will be to interventions that have the potential to lessen that pain. We also hypothesized that psychological factors, such as pain catastrophizing and pain-related fear, would be inversely correlated with openness, given that these factors are linked with poorer functional outcomes, while openness appears to play a key role in chronic pain recovery. This study also explored the relationships between youth and caregiver self-reported openness to interventions. We hypothesized that youth openness and caregiver-expressed openness to each intervention would be strongly correlated with each other, consistent with the extant literature on behavioral modeling influencing the transmission of beliefs from caregiver to child. Identifying key factors associated with openness to various headache management recommendations may constitute a first step towards addressing potential barriers for youth adherence. 

## 2. Materials and Methods

### 2.1. Participants and Procedures 

Participants in this study (*n* = 1087) included children and adolescents (between age 8–17) who presented for evaluation at a multidisciplinary headache clinic, as well as one caregiver. The multidisciplinary headache program is located within a tertiary, urban hospital in the Northeastern United States. Eligibility for participation in this study included: (1) pain chronicity of at least three months prior to evaluation; (2) English-speaking; (3) stable medical and psychiatric status (i.e., no active suicidality or need for inpatient intervention for acute psychological or medical care); and (4) without moderate to severe developmental delay. 

Prior to their initial multidisciplinary headache clinic evaluation, youth and caregivers are routinely asked to complete electronic standardized survey measures via a web-based platform, including data about pain, developmental and medical history, and psychological functioning that are routinely collected as a part of each patient’s standard new-patient multidisciplinary evaluation. These clinical data are then exported to and stored in a centralized data repository [18]. This Chronic Pain Data Repository serves as a database of information on patients seen in the institution’s pain-related clinical settings, governed by a standardized research protocol that has been approved by the hospital’s institutional review board (IRB) since October 2018. De-identified data can then be accessed for specific research studies following a formal application that includes a data safety and monitoring plan, scientific review committee approval, and data use agreement. Importantly, data collected in the web-based platform are part of the clinical standard of care. While youth and families are strongly encouraged to complete the surveys, it is not obligatory, and patients receive all indicated medical treatment irrespective of survey completion status. 

### 2.2. Measures 

Demographic and medical information for youth (i.e., youth age, gender, and race/ethnicity; pain intensity; pain frequency; number of interventions trialed prior to evaluation), and caregivers (i.e., caregiver role, education level, marital status, and employment status) was collected via the clinic’s electronic standardized survey measures. 

### 2.3. Youth-Report Measures 

The Openness to Headache Treatment in Youth (OHT-Y) assesses youth self-reported openness to 15 common interventions typically offered within or recommended by tertiary multidisciplinary headache clinics. This questionnaire was developed by our program for clinical purposes to gauge youth openness to/perceived acceptability of the identified interventions. Each item presents a single headache intervention (e.g., Headache Medication, Dietary Supplements, Psychological Counseling). Responses are rated on a five-point scale, ranging from “Never would consider” to “Definitely would consider.” Ratings are totaled to compute a total score, with higher scores reflecting greater openness to the offered headache interventions. 

The Functional Disability Inventory (FDI) assesses youth appraisal of how disabled they are physically and psychosocially by their chronic pain [19,20]. The inventory consists of 15 items concerning how limited youth feel when engaging in various everyday activities (e.g., walking upstairs, completing chores at home) over the past two weeks. Responses are rated on a five-point scale, ranging from “No trouble” to “Impossible.” Ratings are totaled to compute a total score, with higher scores indicating greater disability in everyday tasks. 

The Headache Impact Test-6 (HIT-6) measures youth perception of the specific impact of headache on daily life over a four-week period [21]. The six items pertain to pain severity, physical function, participation, and mood. Responses are rated on a five-option Likert-type scale ranging from “Never” to “Always.” Ratings are totaled to compute a total score, with higher scores indicating a greater impact of headache pain on daily life. For context, disability levels are defined as: little to no impact (36–49); some impact (50–55); substantial impact (56–59); very severe impact (59–78). 

The Fear of Pain Questionnaire-Child (FOPQ-C) assesses youth pain-related fears and associated avoidance behaviors. Items assess how youth may feel/behave in the context of their pain [22]. Responses are rated on a five-point Likert-type scale, ranging from “Strongly disagree” to “Strongly agree.” Ratings are totaled to compute a total score, with higher scores indicating increased pain-related fear and avoidance. 

The Pain Catastrophizing Scale-Child (PCS-C) assesses youth negative thinking associated with a child’s pain [23]. The 13 PCS-C items assess maladaptive thinking patterns associated with pain. Responses are rated on a five-point Likert-type scale, ranging from “Not at all true” to “Very true.” Ratings are totaled to compute a total score, with higher scores indicating increased catastrophic thinking. 

### 2.4. Caregiver-Report Measures 

The Openness to Headache Treatment in Parents (OHT-P) assesses caregiver self-reported openness to the same 15 interventions assessed in the youth version of the questionnaire. In the same way as the OHT-Y, this measure was developed for clinical purposes to gauge caregiver openness to/perceived acceptability of interventions typically offered within or recommended by tertiary multidisciplinary headache clinics. As with the OHT-Y, responses are rated on a five-point scale, ranging from “Never would consider” to “Definitely would consider.” Ratings are totaled to compute a total score, with higher scores reflecting greater openness to the offered headache interventions. 

The Parent Fear of Pain Questionnaire (PFOPQ) assesses caregiver pain-related fears and associated avoidance behaviors [13]. Items assess how a caregiver may think/behave in the context of their child’s pain. Responses are rated on a five-point Likert-type scale, ranging from “Strongly disagree” to “Strongly agree.” Ratings are totaled to compute a total score, with higher scores indicating increased caregiver pain-related fear and avoidance.

The Pain Catastrophizing Scale-Parent (PCS-P) assesses caregiver negative thinking associated with their child’s pain [24]. Responses are rated on a five-point Likert-type scale, ranging from “Not at all true” to “Very true.” Ratings are totaled to compute a total score, with higher scores indicating increased caregiver catastrophic thinking. 

### 2.5. Statistical Analysis 

Data were analyzed with SPSS version 27. Before conducting the main analyses, the data were tested for normality, linearity, and the presence of outliers. To determine if there were differences in youth gender and race/ethnicity on youth/caregiver openness, independent-samples *t*-tests were conducted. Youth gender was explored as an artificially dichotomous variable (i.e., female vs. male), as <0.5% of this sample endorsed another gender. To determine if there were differences in youth/caregiver openness based on race/ethnicity, caregiver education level, and/or caregiver employment status, univariate analyses of variance (ANOVAs) were conducted. 

Analyses of the hypothesized relationships between openness and key demographic, pain-related, and psychological variables were undertaken, using Pearson correlations to analyze all bivariate relationships. A Bonferroni correction was applied to account for multiple comparisons and significance was evaluated against an alpha level of *p* < 0.003. Finally, to establish the relative importance of the hypothesized demographic variables, pain-related variables, and psychological variables to youth/caregiver openness, a series of hierarchical regressions were conducted. Demographic, pain-related variables, and psychological variables were blocked and entered at sequential steps, with the order of entry being determined by the hypotheses and preliminary analyses. 

## 3. Results

### 3.1. Sample Characteristics and Descriptive Findings

Of 1087 patients, 71.1% patients were female, 86.1% of caregivers were mothers, and 76.7% of the patient–caregiver dyads identified as White. This distribution is expected among youth who present to tertiary pain treatment or chronic headache clinics. Additional descriptive information about the sample can be found in Table 1. 

Means and standard deviations for all measures are presented in Table 2. Nearly all variables, except for the number of interventions previously trialed, were normally distributed. 

Both youth and their caregivers reported a high degree of openness to the 15 individual interventions. The majority (>50%) of youth reported that they would “probably” or “definitely” consider most interventions, with the exception of trigger point injections, Botox injections, intravenous DHE infusion, reiki, and psychopharmacology. Caregivers were similarly open to the same set of interventions as youth, but the majority of caregivers were also open to reiki. See Table 3 for details regarding openness to individual headache interventions. 

### 3.2. Correlates and Group Differences in Youth and Caregiver Openness

Relationships between youth and caregiver openness to individual headache interventions fell in the moderate-high range, with Pearson correlations from 0.42 to 0.73. There were notably strong correlations between youth and caregiver openness to psychopharmacology (0.73), intravenous DHE infusion (0.68), Botox injections (0.67), and trigger point injections (0.66), whereas agreement on openness to dietary recommendations (0.42), sleep hygiene (0.45), and exercise (0.49) fell in the moderate range. See Table 4. 

When examining additional factors correlated with youth/caregiver openness to headache treatments, Pearson correlations (detailed in Table 5) revealed modest but significant correlations between youth/caregiver openness and headache frequency, youth functional disability, youth/caregiver pain-related fear and avoidance, and youth/caregiver pain catastrophizing, as well as between caregiver (but not youth) openness and youth age. Pearson correlations also reflected moderate correlations between youth/caregiver openness and number of interventions previously trialed. Headache intensity was not significantly correlated with youth or caregiver openness. 

Independent samples *t*-tests showed significant differences for youth gender in degree of both youth and caregiver openness. Specifically, participants identifying as female endorsed a higher degree of openness than their counterparts identifying as male: *t* (1076) = 4.89, *p* < 0.001). Caregivers of female youth endorsed a higher degree of openness than caregivers of male youth: *t* (1076) = 2.73, *p* < *0*.01). There were no significant group differences in youth or caregiver openness by caregiver employment status. Two ANOVAs revealed significant differences in both youth (F (4, 1086) = 3.98, *p* < 0.01) and caregiver openness (F(4, 1086) = 3.98, *p* < 0.01) dependent upon caregiver education. Specifically, Tukey post-hoc pairwise comparisons indicated that youth of caregivers who reported high-school level education endorsed a lower degree of openness than those reporting higher levels of education. Similarly, those same caregivers reporting high-school level education endorsed a lower degree of their own openness than those with higher levels of education. 

### 3.3. Hierarchical Regressions Predicting Youth and Caregiver Openness 

Hierarchical multiple regression was used to assess the ability of pain-related factors and psychological factors to predict levels of youth/caregiver openness to headache treatment, after controlling for the effects of demographic variables. Preliminary analyses were conducted to ensure no violation of the assumptions of normality, linearity, multicollinearity, and homoscedasticity. For the youth model, gender and caregiver education were entered at Step 1, and explained 2.5% of the variance in youth openness. Upon entering pain-related factors in Step 2, the total variance explained by the model was 12.7%: F(4, 1072) = 40.06, *p* < 0.001. Pain-related factors explained an additional 10.3% of the variance in youth openness. After the entry of psychological factors in Step 3, the total variance explained by the model was 15.8%: F(8, 1068) = 26.33, *p* < 0.001. The psychological factors explained an additional 3.5% of the variance in youth openness. In the final model (Step 4), after the entry of caregiver openness, the total variance explained by the model as a whole was 47.3%: F(9, 1067) = 108.21, *p* < 0.001. Caregiver openness explained 31.2% of the variance in youth openness. In the final model, only youth gender, number of interventions previously trialed, and caregiver openness were statistically significant predictors of youth openness. 

For the caregiver model, youth gender, youth age, and caregiver education were entered at Step 1 and collectively explained 3.1% of the variance in caregiver openness. Upon entering pain-related factors in Step 2, the total variance explained by the model was 13.4%: F(5, 1071) = 33.04, *p* < 0.001. Pain-related factors explained an additional 10.3% of the variance in caregiver openness. After the entry of psychological factors in Step 3, the total variance explained by the model was 15.8%: F(9,1067) = 22.20, *p* < 0.001. The psychological factors explained an additional 2.4% of the variance in caregiver openness. In the final model (Step 4), after the entry of youth openness, the total variance explained by the model as a whole was 47.9%: F(10,1066) = 98.00, *p* < *0*.001. Youth openness explained 32.1% of the variance in caregiver openness. In the final model, only caregiver education, number of interventions previously trialed, overall impact of headaches on youth, and youth openness were statistically significant predictors of caregiver openness. See Table 6 for detailed findings for both hierarchical regression models. 

## 4. Discussion

Despite the prevalence of non-adherence in pediatric chronic pain, research exploring relevant underlying factors is scant. This study sought to explore one potentially relevant factor in the discussion of adherence in pediatric pain management: youth and caregiver expressed openness to interventions. This study is novel in its exploration of contextual variables that might influence patient and caregiver openness to various treatment options prior to initial evaluation in a tertiary pediatric headache clinic. Specifically, we examined the relationships between youth- and caregiver-expressed openness to various headache interventions and relevant demographic factors, pain-related factors, and psychological factors. 

When exploring predictors of youth and caregiver openness to headache interventions, overall, we found that the most prevalent factor in explaining variance in youth openness was caregiver openness and vice versa, accounting for more than 30% of the variance in each case. This finding is unsurprising, given the moderate to high correlations between not only youth and caregiver total openness scores, but also the moderate to high correlations between youth/caregiver openness to each of the individual interventions. These findings are consistent with studies establishing the influence of caregiver modeling of pain attitudes, beliefs, and coping behaviors on development and maintenance of pediatric chronic pain [14,15]. We also found that pain-related fear and pain catastrophizing explained less than 4% of the variance, while pain-related factors (e.g., headache frequency and number of interventions previously trialed) explained over 10% of the variance in both the youth and caregiver openness hierarchical regression models. These results were unexpected, given that pain catastrophizing, pain-related fear and avoidance, and subsequent functional disability are well-established, influential factors in (and oftentimes barriers to) the management of pediatric pain [25]. It may be that the percentage of variance in openness accounted for by these psychological variables was lower than anticipated because of where patients/caregivers are in their pain journey. That is, these constructs were assessed prior to initial multidisciplinary evaluation, whereby anecdotally many families are focused on relaying medical information and seeking medical treatment. Thus, these factors may drive decisions about openness to interventions more than psychological variables. Additionally, such families may be early in the pain–disability cycle, whereby such psychological variables may not be as deeply entrenched so as to drive all pain-related behavior.

In terms of understanding youth openness specifically, we found their gender to be a significant predictor of openness. Female-identifying patients were more likely to report higher openness than male-identifying patients. This finding is well-supported by the extant literature on gender roles, emotional openness, and psychological barriers to help-seeking. For instance, Komiya, Good, and Sherrod found that college women reported significantly more emotional openness and thus were more likely to seek psychological help, as compared to men [26]. Gender differences in socialization around emotions may also be contributing. Cox and Mezulis noted differences in maternal responses to their children, with young girls more likely to be encouraged to express emotions and to make emotion-focused contributions [27]. This could explain at least in part the gender differences in openness to headache interventions. While both males and females were similarly open to wellness-based recommendations (e.g., diet, exercise, sleep hygiene) and cognitive behavioral therapy, female patients were significantly more open to a broader range of headache interventions (e.g., medication, injections, yoga, meditation, aromatherapy) than their male counterparts. 

In terms of understanding caregiver openness specifically, we found caregiver education and youth-perceived impact of the headaches to be a significant predictor of caregiver openness. Youth of caregivers with education beyond high school reported higher openness scores than those with a high-school education. This is supported by Sutin and colleagues, who found that parental educational attainment was strongly correlated with child openness [28]. Though caregiver distress was not explicitly measured in this study, it is possible that witnessing their child’s increased disability and the negative impact of headaches increases distress for caregivers, who in turn are open to anything that could potentially relieve their child’s pain. 

Finally, in predicting both youth and caregiver openness, those who had previously trialed more individual headache interventions also reported increased openness. Given that Woo and colleagues defined openness as a major dimension of one’s personality, it seems reasonable that prior openness to interventions would play a role in predicting future openness [29]. Additionally, if youth and their caregivers have trialed numerous interventions for headache without sustained benefit (which is often the case for patients being evaluated in a specialized, multidisciplinary headache clinic), they may report increased openness to a wider variety of interventions than those who are just starting to explore headache management. In other words, those who have already trialed numerous interventions may be open to “anything that could help.” 

### 4.1. Limitations of the Present Study 

There are various limitations associated with this study, beginning with the homogeneous nature of the sample (71.1% female-identifying, 76.7% White/non-Hispanic, 86.1% mothers as caregiver). This reduces the generalizability of these findings to more diverse patient populations. Future studies should seek to replicate these findings in more diverse samples, wherein race/ethnicity can be appropriately included in analyses. Another limitation of this study is the exclusive use of self-report questionnaires, which may be skewed due to reporter bias. Extending this, although caregivers and youth were sent separate questionnaires to complete prior to the clinic evaluation, it is not known whether the caregiver and youth completed their questionnaires independently, simultaneously, or collaboratively. Thus, it is unclear the extent to which youth responses were more directly influenced by a caregiver at the time of completion. Future studies could consider requesting that participants complete their questionnaires in the clinic to ensure independent reporting. 

Additionally, data in this study were collected prior to the initial clinic evaluation. It is certainly possible that, as a result of the education provided during the evaluation and while presenting treatment recommendations, youth and caregiver openness might change, especially when a description and rationale for various interventions are discussed in detail. Indeed, in this study, youth and caregivers were provided with the name of the intervention only, so their self-reported openness may be influenced with familiarity with the intervention. Therefore, future studies, particularly those that wish to connect openness more directly to adherence, might consider assessing openness immediately following the clinic visit to account for potential interim changes. Relatedly, as this was a cross-sectional study, examining variables at a single point in time, we did not collect information on patient adherence to the specific interventions recommended during the clinic evaluation. This is the clearest future direction for this study, given the potential for relationships between openness and treatment adherence. Furthermore, there are numerous additional factors that may play a role in openness to intervention, many of which are key social determinants of health (e.g., access to health care, financial resources). Our study did not have access to data (self-reported or otherwise) regarding these critical factors and they will be important to include in future studies of openness and adherence. 

Finally, the scope of this study was limited to youth with chronic headache disorders. Therefore, our results cannot yet be generalized to other chronic pain presentations, though this could be a promising focus of future study. We were also not able to analyze the effect of headache diagnosis on openness as diagnostic data were unavailable at the time of the study. Future studies could explore whether specific headache diagnosis (i.e., chronic migraine, new daily persistent headache, tension-type headache, etc.) is an influential factor in youth and caregiver openness to intervention. 

### 4.2. Clinical Implications 

Our results highlight the importance of considering additional perspectives within the family system as crucial context for understanding youth and caregiver openness, and potentially by extension, adherence. This is consistent with Caruso and colleagues, who found that mothers who sought their child’s input and offered choice in the medical decision-making process had more adherent children [30]. Indeed, as McGrath and colleagues suggested, caregivers are often the “gatekeepers” to children’s healthcare [31]. This suggests that concordance between caregiver and patient perspectives (and the ability to partake in nuanced conversations to achieve such concordance) could be a target for therapeutic intervention, as a means of first increasing openness to interventions and increasing the likelihood of adherence to treatment recommendations. 

Having an awareness of patient and caregiver beliefs (including preconceived notions) about typically offered treatment options may also inform the ways in which clinicians educate families about various pain interventions during their visit. Specifically, taking a precursory inventory of patient and caregiver beliefs offers an opportunity for clinicians to personalize healthcare. Doing so empowers providers with context about their patients, enabling them to tailor conversations in clinic to each family. This exercise may also be a helpful tool in building rapport between patient and healthcare provider, while simultaneously focusing the conversation during time-limited appointments on the treatment modalities the family is most open to. 

Lastly, understanding youth and caregiver openness may also streamline the overall headache management experience. For many, headache treatment is characterized by trial-and-error, a lengthy process of trying many interventions, often complicated by side effects and feelings of frustration and hopelessness. Perhaps by collaborating with families to strategically choose initial options that align well with their belief system, improved rapport and buy-in will increase treatment engagement, therein holding potential for quicker recovery. In fact, future research could examine how patient openness to the recommended treatment modality influences treatment outcomes. 

## 5. Conclusions

Pediatric chronic headaches are highly prevalent, challenging to treat, and often complicated by difficulties with treatment adherence. This study sought to gain insight into a rather unexplored factor that may underlie adherence: openness to intervention. Specifically, we sought to explore what demographic, pain-related, and psychological factors influence youth and caregiver openness to various headache interventions. Our findings highlight the importance of involving caregiver perspectives in pediatric headache management, as, far and away, the most significant predictor of youth and caregiver openness to intervention was their counterpart’s openness to the same. It is hoped that a better understanding patient and caregiver openness to treatment options can provide insight on joining with families, meeting them where they are, and overcoming barriers to adherence. If so, this foundational knowledge has the potential to help ameliorate the burden of pediatric chronic headaches on our patients, families, communities, schools, and the healthcare system.

## Figures and Tables

**Table 1 children-09-01956-t001:** Youth/Caregiver Characteristics and Descriptive Data.

	*N* (%)	M (SD)
Youth Self-Reported Gender		
Female	773 (71.1%)	
Male	305 (28.1%)	
Non-Binary	5 (0.5%)	
Not reported	4 (0.4%)	
Youth Race/Ethnicity		
White, non-Hispanic	834 (76.7%)	
Hispanic	36 (3.3%)	
African American/Black, non-Hispanic	15 (1.4%)	
Asian, non-Hispanic	11 (1.0%)	
Another race, non-Hispanic	27 (2.5%)	
Multiracial, non-Hispanic	9 (0.8%)	
Not reported	155 (14.3%)	
Youth Age, years	-	14.45 (2.42)
Caregiver Relationship with Youth		
Biological Mother	936 (86.1%)	
Biological Father	113 (10.4%)	
Adoptive Mother	27 (2.5%)	
Adoptive Father	4 (0.4%)	
Other (e.g., Guardian)	7 (0.7%)	
Caregiver Education Level		
High School	125 (11.5%)	
Some College	128 (11.8%)	
Bachelor’s Degree	380 (35.0%)	
Master’s Degree	335 (30.8%)	
Doctoral Degree	119 (10.9%)	
Caregiver Employment Status		
Not Employed	53 (4.9%)	
Disabled	18 (1.7%)	
Homemaker	123 (11.4%)	
Part-time Employment	232 (21.5%)	
Full-time Employment	623 (57.8%)	
Other	28 (2.6%)	

**Table 2 children-09-01956-t002:** Descriptive Statistics for all Study Measures.

	*N* (%)	M (SD)
Typical Headache Pain Intensity (NRS; 0–10)	-	7.41 (1.69)
Headache Frequency (per week)		-
Less than once a week	93 (8.6%)	
Once a week	67 (6.2%)	
A few days per week	269 (24.8%)	
Daily, but not constant	133 (12.2%)	
All day, every day (constant)	524 (48.3%)	
Number of Interventions Trialed	-	1.76 (1.87)
Youth Openness Total (OHT-Y; 0–60)	-	38.51 (11.09)
Caregiver Openness Total (OHT-P; 0–60)	-	43.29 (10.09)
Functional Disability (FDI; 0–60)	-	13.10 (11.03)
Headache Impact Test (HIT-6; 36–78)	-	63.49 (6.92)
Fear of Pain-Child (FOPQ-C; 0–96)	-	39.34 (19.18)
Pain Catastrophizing-Child (PCS-C; 0–52)	-	22.74 (12.41)
Fear of Pain-Parent (PFOPQ; 0–96)	-	27.43 (14.65)
Pain Catastrophizing-Parent (PCS-P; 0–52)	-	16.44 (10.72)

**Table 3 children-09-01956-t003:** Youth/Caregiver Openness to Individual Headache Interventions.

Item	% Endorsing Definitely Would NOT Consider (0)		% Endorsing Probably Would NOT Consider (1)		% Endorsing May or May Not Consider (2)		% Endorsing Probably Would Consider (3)		% Endorsing Definitely Would Consider (4)	
	Youth	CG	Youth	CG	Youth	CG	Youth	CG	Youth	CG
1. Headache medication	1.1	0.8	4.0	3.0	13.2	14.0	22.1	22.2	56.9	60.0
2. Psychological Counseling	6.3	2.5	12.4	6.3	22.7	19.1	24.6	22.8	33.9	49.3
3. Cognitive-Behavioral Therapy	8.6	3.2	13.4	6.7	27.9	22.8	24.7	25.0	25.4	42.2
4. Psychopharmacology	13.9	8.3	18.2	17.6	24.0	28.3	18.1	17.3	25.8	28.5
5. Trigger-point injections	21.6	13.5	21.2	18.4	29.3	31.5	15.6	19.3	12.3	17.3
6. Botox injections	32.5	20.4	20.8	19.5	26.2	31.1	10.1	14.3	10.4	14.7
7. Intravenous DHE infusion	15.6	13.2	19.3	18.5	30.0	32.3	17.5	18.8	17.6	17.2
8. Acupuncture	15.0	7.0	12.6	10.0	21.6	21.2	21.7	23.5	29.1	38.4
9. Reiki	12.7	7.0	14.0	8.9	24.7	22.0	23.1	25.0	25.6	37.1
10. Yoga	7.0	2.9	10.7	4.8	20.3	16.1	26.2	22.9	35.8	53.3
11. Aromatherapy	8.7	4.0	11.5	6.4	21.0	16.7	23.6	23.9	35.1	48.9
12. Meditation	5.6	2.3	11.1	5.1	21.3	17.8	27.0	24.8	35.0	50.0
13. Dietary Recommendations	2.3	0.5	4.1	0.8	17.2	9.5	31.0	27.7	45.4	61.5
14. Sleep Hygiene	3.0	1.6	3.7	1.6	14.4	11.9	27.6	22.0	51.3	63.0
15. Exercise	1.4	0.5	2.7	1.0	11.5	7.6	24.7	20.5	59.8	70.4

**Table 4 children-09-01956-t004:** Youth and Caregiver Openness Correlation Coefficients for Individual OHT Items and Total Score.

Item	*r*
Headache medication	0.56 **
2. Psychological Counseling	0.62 **
3. Cognitive-Behavioral Therapy	0.57 **
4. Psychopharmacology	0.73 **
5. Trigger-point injections	0.66 **
6. Botox injections	0.67 **
7. Intravenous DHE infusion	0.68 **
8. Acupuncture	0.65 **
9. Reiki	0.65 **
10. Yoga	0.58 **
11. Aromatherapy	0.60 **
12. Meditation	0.55 **
13. Dietary Recommendations	0.42 **
14. Sleep Hygiene	0.45 **
15. Exercise	0.49 **
OHT Total Score	0.67 **

Note: ** *p* < 0.003, with applied Bonferroni correction (*p* < 0.05/15 = *p* < 0.003).

**Table 5 children-09-01956-t005:** Correlation Matrix of between Youth/Caregiver Openness and Pain-Related/Psychological Factors.

Variable	Youth Openness (OHT-Y)	Caregiver Openness (OHT-P)
Demographic Factors		
Youth Age	*n.s.*	0.10 **
Pain-Related Factors		
Headache Intensity	*n.s.*	*n.s.*
Headache Frequency	0.17 **	0.18 **
# Interventions Trialed	0.34 **	0.34 **
Psychological Factors		
FDI-C	0.21 **	0.20 **
HIT-6	0.21 **	0.22 **
FOPQ-C	0.24 **	-
PCS-C	0.21 **	-
PFOPQ	-	0.09 **
PCS-P	-	0.07 *

Note: * *p* < 0.05, ** *p* < 0.01., *n.s.* = not significant.

**Table 6 children-09-01956-t006:** Hierarchical Regressions Predicting Youth and Caregiver Openness.

Variable	B	*β*	*sr*	Δ*R*^2^	Adjusted *R*^2^
Youth Model					
Step 1				0.03 ***	0.03
Youth Gender	−1.97	−0.08	−0.11 ***		
Caregiver Education	−0.05	−0.01	−0.01		
Step 2				0.10 ***	0.13
Headache Frequency	0.01	0.00	0.00		
# Interventions Trialed	0.63	0.11	0.13 ***		
Step 3				0.04 ***	0.16
FDI	−0.00	−0.00	−0.00		
HIT	−0.01	−0.01	−0.01		
FOPQ-C	0.03	0.05	0.04		
PCS-C	0.04	0.04	0.04		
Step 4				0.31 ***	0.47
OHT-P	0.67	0.61	0.61 ***		
Caregiver Model					
Step 1				0.03 ***	0.03
Youth Age	0.15	0.04	0.05		
Youth Gender	0.95	0.04	0.06		
Caregiver Education	0.80	0.09	0.12 ***		
Step 2				0.10 ***	0.13
Headache Frequency	0.23	0.03	0.04		
# Interventions Trialed	0.52	0.10	0.12 ***		
Step 3				0.02 ***	0.15
FDI	0.02	0.02	0.02		
HIT-6	0.10	0.07	0.08 *		
PFOPQ	−0.02	−0.03	−0.02		
PCS-P	0.01	0.01	0.01		
Step 4				0.32 ***	0.47
OHT-Y	0.56	0.62	0.62 ***		

Note: * *p* < 0.05, *** *p* < 0.001.

## Data Availability

The data presented in this study are available on request from the 369 corresponding author. The data are not publicly available due to privacy.

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
