# Peer review of "Aligning with Patients and Families: Exploring Youth and Caregiver Openness to Pediatric Headache Interventions"

_children, 2022, doi:10.3390/children9121956_

Round 1

Reviewer 1 Report

Thank you for submitting the manuscript. I have read your manuscript with great interest. The topic you are discussing is very interesting and important. It is always difficult to deal with chronic pain, both from a clinical and research point of view, especially in the pediatric field. It is difficult to read a manuscript like yours: clear, well structured, reasoned and of quality. Therefore, even trying to find some critical element, I didn't succeed. I just have to congratulate you.

Kind Regards

Author Response

We appreciate the kind feedback from Reviewer 1. 

Reviewer 2 Report

This report by Allison M. Smith et al explored the role of the Youth and their Caregiver Openness to various headache interventions and their relationships with various demographic and psychological factors. One of the study’s highlights is the openness of caregivers in pediatric headache management, which influences youths' openness and involvement in treatment adherence.

Overall the manuscript is written well and the results are well supported by the data. I have no further comments and the manuscript is acceptable for publication.

Author Response

We appreciate the kind feedback from Reviewer 2.

Reviewer 3 Report

This manuscript describes the evaluation of openness of youth and caregivers to pediatric headache interventions. In my opinion this paper is logically well written and is well described. I have a few minor comments to this paper.

1) Results: I would like the authors to start each result subsection with a heading. That way it becomes very convenient for the reader to understand what the paragraph below is discussing. It was a little confusing to jump from one result to another without headings.

2) Have the authors observed if any of the participants became more adherent to treatment if they were convinced by their physician or a physician when brought to the pain center since the questionnaire may not be accurately filled by the patient or caregiver?

3) Have the authors looked at other factors of non-adherence to treatment by different groups such as access to headache/pain care or financial status? Does that in any way influence this study?

Author Response

We appreciate the cogent suggestions from Reviewer 3.

  • Results: I would like the authors to start each result subsection with a heading. That way it becomes very convenient for the reader to understand what the paragraph below is discussing. It was a little confusing to jump from one result to another without headings.
    1. Descriptive headings have been added to the Results for enhanced clarity on p. 5-9.
  • Have the authors observed if any of the participants became more adherent to treatment if they were convinced by their physician or a physician when brought to the pain center since the questionnaire may not be accurately filled by the patient or caregiver?
    1. Because our study did not evaluate adherence to specific treatment recommendations, we do not have the data to address this question. However, because the openness to interventions was assessed prior to the initial visit, it is indeed possible that openness AND likelihood of adherence would change after discussions with health care provider during the visit. We discuss these implications on p. 11 lines 477-488 of the Discussion section.
  • Have the authors looked at other factors of non-adherence to treatment by different groups such as access to headache/pain care or financial status? Does that in any way influence this study?
    1. Our study did not collect/have access to data (self-reported or otherwise) regarding access to care and/or financial status of the participants and their families. However, we agree that these are likelihood key factors that influence non-adherence, in addition to openness and important to include in future studies of adherence. We have added commentary to this effect on p. 11 lines 489-493 of the Discussion section.

Reviewer 4 Report

Overall, this manuscript describes the openness of youth and caregiver to pediatric headache interventions, and the results are consistent with other studies. Some results (and therefore the conclusions) may not be supported by their data. For example, In line 258, they claimed that “There were no significant group differences in youth openness by race/ethnicity or caregiver employment status”, however, the race/ethnicity of the subjects is predominantly white (76.7%) with 14.3% as non-reported. Therefore, it is not appropriate to include race in the analysis. The analysis of the relationship between openness with the caregiver’s education also needs to be careful, as the majority of caregivers with some college-level education, and only 11.5% with high school education. The authors discussed the limitations of the imbalanced population in the study. For the statistical analysis, however, did they try to adjust the imbalanced population? If so, they should mention the adjustment in the method section, and if not, they need to discuss why they did not do the adjustment. It is also interesting to see that from this study, pain-related fear and pain catastrophizing explained less than 4% of the variance, while pain-related factors (e.g., headache frequency and the number of interventions previously trialed) explained over 10% of the variance. Did authors have any explanation for this? The authors should discuss this further.

Minor issues: Some descriptions are redundant. For example, lines 249 to line 251 on the correlation of youth openness and interventions previously trialed is almost the same as lines 263-265. They may consolidate some of those descriptions (not just limited to the one mentioned above) through the manuscript.

Author Response

We appreciate the cogent suggestions from Reviewer 4.

  • In line 258, they claimed that “There were no significant group differences in youth openness by race/ethnicity or caregiver employment status”, however, the race/ethnicity of the subjects is predominantly white (76.7%) with 14.3% as non-reported. Therefore, it is not appropriate to include race in the analysis.
    1. We agree with this point and have removed race from the analyses.
  • The analysis of the relationship between openness with the caregiver’s education also needs to be careful, as the majority of caregivers with some college-level education, and only 11.5% with high school education. The authors discussed the limitations of the imbalanced population in the study. For the statistical analysis, however, did they try to adjust the imbalanced population? If so, they should mention the adjustment in the method section, and if not, they need to discuss why they did not do the adjustment.
    1. We worry that the reviewer may have inadvertently be misled by the analyses related to caregiver education. This variable is NOT nor was it treated as a dichotomous variable. It is an ordinal variable (see descriptives in Table 1) and one of just two categorical variables included in the regression analyses. To further clarify, the identification of group differences via ANOVAs on p. 8 included post-hoc Tukey tests for pairwise comparisons, to understand specifically where those differences were. This has been clarified in the text. Thus, with this, we kept caregiver education in our analyses.
  • It is also interesting to see that from this study, pain-related fear and pain catastrophizing explained less than 4% of the variance, while pain-related factors (e.g., headache frequency and the number of interventions previously trialed) explained over 10% of the variance. Did authors have any explanation for this? The authors should discuss this further.
    1. We agree that this was an interesting finding. We offer our hypothesized explanation for this on p. 10 lines 419-427 in the Discussion section (enhanced in this revision).
  • Some descriptions are redundant. For example, lines 249 to line 251 on the correlation of youth openness and interventions previously trialed is almost the same as lines 263-265. They may consolidate some of those descriptions (not just limited to the one mentioned above) through the manuscript.
    1. Redundant descriptions have been consolidated throughout the Results section, as requested, mostly notably on the full re-write at the top of p. 8 (lines 254-274). We were unable to streamline the discussion of the hierarchical regressions on p. 9, as the models were too disparate to combine discussion without obscuring the findings.

Round 2

Reviewer 4 Report

I am fine with the response from authors.